# Application of Mathematical Models to Determine the Feasibility of Amorphous Drug Layering in Pan Coaters

**DOI:** 10.3390/pharmaceutics14010149

**Published:** 2022-01-08

**Authors:** Michael Choi, Stuart C. Porter, Axel Meisen

**Affiliations:** 1Particle Coating Technology Solutions, Inc., Lyndell, PA 19354, USA; 2PPT Pharm Technologies, Hatfield, PA 19440, USA; sporterpt@aol.com; 3Department of Chemical and Biological Engineering, The University of British Columbia, Vancouver, BC V6T 1Z3, Canada; axel@meisen.ca

**Keywords:** drug-layering, accuracy, precision, coating efficiency, coating uniformity, acceptance value, process capability

## Abstract

Oral solid dosage forms that contain APIs in the amorphous state have become commonplace because of many drug substances exhibiting poor water solubility, which negatively impacts their absorption in the human GI tract. While micronization, solvent spray-drying, and hot-melt extrusion can address solubility issues, spray coating of the APIs onto beads and tablets offers another option for producing amorphous drug products. High-level comparisons between bead and tablet coating technologies have the potential for simpler equipment and operation that can reduce the cost of development and manufacturing. However, spray coating directly onto tablets is not without challenges, especially with respect to meeting uniformity acceptance value (*AV*) criteria, comprising accuracy (mean) and precision (variance) objectives. The feasibility of meeting *AV* criteria is examined, based on mathematical models for accuracy and precision. The results indicate that the main difficulty in manufacturing satisfactory drug-layered tablets by spray coating is caused by the practical limitations of achieving the necessary coating precision. Despite this limitation, it is shown that *AV* criteria can be consistently met by appropriate materials monitoring and control as well as processing equipment setup, operation, and maintenance.

## 1. Introduction

The main advantage of creating oral solid dosage forms containing an amorphous active pharmaceutical ingredient (API) is the improved drug solubility in the human GI tract that leads to higher bioavailability. Micronizing, solvent spray drying, and hot-melt extrusion are classic techniques for creating APIs with enhanced solubility characteristics, but these techniques involve relatively complex procedures with high manufacturing costs. A recent study on the feasibility of coating an amorphous solid dispersion onto beads [1] confirms the feasibility of active drug-layering directly onto pellets or tablets, with the benefits of reducing complexity as well as material and manufacturing costs. Drug products with drug-layered pellets have a historical significance [2] and, as shown in Table 1, the manufacturing process requires fewer steps compared to those for typical solid-dosage forms prepared, for example, when using amorphous APIs obtained using the classic techniques. This table also indicates the potential reduction in the number of manufacturing unit operations required to produce the final dosage form, which can be of considerable benefit not only for development cost and time-to-market but also for commercial manufacturing when considering the capital and operating costs of multiple unit operations. 

Regardless of which technique is used to prepare APIs with enhanced aqueous solubilities, all of them have one attribute in common, i.e., the API is combined with polymers having good aqueous solubilities under physiological conditions. Traditionally, the polymers used have been highly water soluble, such as polyethylene glycol, hypromellose, polyvinylpyrrolidone, and polyvinylpyrrolidone–vinyl acetate copolymers. One of the potential disadvantages of these polymers is that the stomach—the first main compartment of the human GI tract to which the dosage form is exposed—is not a primary absorption site for drug substances. The aqueous nature of the stomach contents can cause the amorphous solid dispersion to release the API, which may immediately begin to recrystallize (and hence become insoluble again) before the API reaches a primary absorption site in the upper part of the small intestine. To overcome this problem, interest in delayed-release polymers (such as hydroxypropyl methylcellulose acetate succinate, HPMCAS, and hydroxypropyl methylcellulose phthalate, HPMCP) has grown. These polymers allow the amorphous solid dispersion to remain intact until it reaches the upper part of the small intestine, where the polymers dissolve. This approach is also valid for drug-layering using coating techniques.

For solubility enhancement techniques using spray drying and drug-layering processes, the API is typically combined with the selected polymer to form a solution by means of a common solvent.

For over the counter (OTC) drugs, tablet formulations are often preferred to capsule formulations since they are less prone to product tampering and counterfeiting. For this market segment, pellets are therefore often compressed into tablets—also known as tablets of multi-unit pellet system (MUPS) [3]. An alternative to MUPS involves the direct application of the API onto tablets (herein called DCT, short for ‘Drug Coated Tablets’ to differentiate them from MUPS tablets) that can reduce the number of typical manufacturing unit operations using micronization of API or MUPS by 67-75% (see Table 1). Historically, drug-layering on tablets has typically been utilized for more complex formulations where the drug’s coating is used as the immediate-release dose and the drug in the core is used as the sustained-release dose. For this reason and the difficulty in controlling the coating uniformity to meet the drug content uniformity requirement, drug-layering in pan coaters can be considered a more complicated and challenging process.

While drug-layering onto pellets is typically performed using fluid-bed (Wurster) techniques, pan coating is more prominently used for tablets [4,5,6]. Pellet systems are inherently less susceptible to coating uniformity issues since the final dose comprises many pellets so that coating variability is effectively evened out. An important challenge with pan-coating technology, including its use to produce DCT, is its compliance with the content uniformity criteria, specifically the acceptance value. In this paper, recent tablet coating models [7,8] are used to show that it is possible to meet the criteria by a judicious choice of design and operating parameters.

## 2. Methods

The methods used to derive the relationship between the acceptance value criteria, accuracy, and precision parameters are provided in this section. However, since applying amorphous drugs directly onto tablets in pan coaters is a relatively new concept, example approaches for preparing tablets and the coating solution, and for processing to ensure the amorphous state of API, are provided first.

### 2.1. Description of Amorphous Drug Coating Process in Pan Coaters

Before the application of a coating solution containing an API to a suitable tablet core can be considered, key development steps must be undertaken, namely:Selection of a suitable polymer that is compatible with the API, which can maintain the amorphous state of the API during application of the coating to the tablets, and ensure that once the final dosage form is consumed by the patient, the amorphous nature of the API is maintained right up to the delivery to the primary site of absorption.Selection of a suitable solvent system that is capable of dissolving both the polymer and the API. Ideally, the solvent should be a single component system. If a solvent mixture is selected, ideally, a constant boiling (i.e., an azeotrope) mixture should be used or one where the least volatile solvent in the mixture can maintain both the API and polymer in the solution.Design of an optimized coating process that allows the coating to be uniformly deposited on the substrate, and allows for the control of the drying process, so that once a dry coating is formed on the substrate, the API still retains its amorphous characteristics.

Once the necessary developmental work has been accomplished, the application process typically comprises the following steps:Creating a coating solution by dissolving the polymer and the API in the requisite solvent system;Loading the tablet cores into the coating pan;Warming the tablets until they reach the required temperature;Coating the tablets by using the optimal coating process conditions described below;Drying the coated tablets at the conclusion of the coating process to remove traces of residual solvent(s);Cooling the tablets;Emptying the coating pan.

Modern pan coaters typically have cylindrical rotary drums with tapered ends and baffles to turnover and mix the bed of tablets efficiently [9]. A coating solution containing amorphous API is sprayed onto the bed surface with the aid of pneumatic nozzles located along the bed axis and utilizing patterned air to enhance the spray coverage over the bed surface. The coating substrate remains on the tablets as the solvent in the coating solution is removed by conditioned processed air that is drawn through the bed of tablets and perforated drum walls. Repeated tablet passages through the spray zone, caused by the rotation of the pan and aided by baffles, improve the coating uniformity on the surface and between the tablets. The process is concluded when the specified quantity of amorphous API is applied on the tablets that meet the content uniformity acceptance value.

### 2.2. Acceptance Value and Its Relationship to Accuracy and Precision in Pan Coating Processes

The API content uniformity acceptance value (*AV*) is defined in the harmonized USP/EP/JP Convention [10]:(1)$$AV=\left|M-X\right|+k_{AV}s$$
where *M* is the reference value, *X* is the mean of the individual API content as a percentage of the label claim, *k_AV_* is the acceptability constant (*k_AV_* = 2.4 and 2.0 for sample sizes (*n_s_*) of 10 and 30, respectively), and *s* is the sample standard deviation. For drug-layering processes, *X* and *s* represent the average content and variability of the API in the coating, respectively. The content uniformity criterion is met if *AV* is equal to or less than 15 for the level 1 test and 25 for the level 2 test. Equation (1) is also conditional on the value of *X*:(2)$$AV=k_{AV}s\;\mathrm{for}\;98.5\%\;\leq \;X\;\leq \;101.5\%$$
(3)$$AV=98.5-X+k_{AV}s\;\mathrm{for}\;X\;<\;98.5\%$$
(4)$$AV=X-101.5+k_{AV}s\;\mathrm{for}\;X\;>\;101.5\%$$

USP/EP/JP further specifies that, for a target content per dosage unit (*T*) at the time of manufacture greater than 101.5%, Equations (2) and (4) must be replaced with
(5)$$AV=k_{AV}s\;\mathrm{for}\;98.5\%\;\leq \;X\;\leq \;T$$
(6)$$AV=X-T+k_{AV}s\;\mathrm{for}\;X\;>\;T$$

The above relationships between *s* and *X* are shown in Figure 1 for the level 1 criterion (i.e., *AV* ≤ 15) for *n_s_* = 10 and 30, and *T* = 100 and 102.5%. Any combination of *X* and *s* that falls within the green-colored region under the curves represents acceptable operating ranges of *AV* ≤ 15. For example, when *n_s_* = 10 and *T* = 100%, *s* must be equal to or less than 6.25% to meet this *AV* criterion for *X* between 98.5 and 101.5%. However, if *X* falls outside this range, the penalty comes in the form of a lower acceptable *s*, which decreases approximately proportional to the proximity of the 98.5–101.5% range, as shown in Figure 1. For example, if the process capability is such that *X* = 97.0%, *s* must be less than 5.83% to meet the level 1 *AV* criterion (compared to 6.25% at 98.5% ≤ *X* ≤ 101.5%). Consequently, tighter control on the coating variability (i.e., *s*) is needed when *X* deviates from the 98.5–101.5% range. Increasing *n_s_* to 30 increases the acceptable operating range of coating variability (shown in yellow), resulting in *s* being equal to or less than 7.5% if *X* is between 98.5 and 101.5%, and increasing *T* to 102.5% widens the flat operating range to 98.5 and 102.5%, as shown by the blue and orange regions for *n_s_* = 10 and 30, respectively, in Figure 1.

Coating uniformity is commonly characterized in terms of the coefficient of variation (*CV*) based on the entire population, which is the true measure of the process variability, and *CV* is widely used in modeling coating uniformity [11]. Hence, *s* may be converted to *CV* using the relationship
(7)$$s=\sqrt{\frac{n_{s}}{n_{s}-1}}X\cdot CV$$

Substituting Equation (5) into (1) gives
(8)$$AV=\left|M-X\right|+k_{AV}^{\prime }X\cdot CV$$
(9)$$\mathrm{where}\;k_{AV}^{\prime }=k_{AV}\sqrt{\frac{n_{s}}{n_{s}-1}}$$

Statistically, *X* denotes the ‘accuracy’ of the process since it is the measure of closeness to the target, and *CV* denotes the ‘precision’ of the process since it is the measure of the variability of the API content in the tablet coatings.

## 3. Results and Discussion

Figure 2, using the same color schemes for the values of *n_s_* and *T* as Figure 1, shows the relationship between the acceptance value, accuracy, and precision, as described by Equation (8). This relationship indicates that *CV* values must be less than *s* to satisfy the acceptance value criterion. For example, *CV* must be less than 6% (see Figure 2) compared with *s* less than 6.25% at *X* = *T* = 100% (see Figure 1) to satisfy *AV* ≤ 15. This simply means that the true process variability must be lower than 6% to meet the acceptance criterion for *AV* for 98.5 ≤ *X* ≤ 101.5% and *T* ≤ 101.5%. If the process capability of *X* is outside the 98.5–101.5% range but within 95–105%, *CV* must be controlled to no higher than 4% for *T* ≤ 101.5% to meet the acceptance criterion for *AV*. One way of improving the *CV* from 6% to 4% is to increase the coating time (e.g., by decreasing the spray rate or solid’s concentration in the coating solution), but this can be costly from an operational perspective—i.e., to decrease *CV* from 6 to 4% requires the coating time to be more than doubled based on the inverse square root relationship between *CV* and coating time [4,7,12,13]. Increasing *n_s_* to 30 can increase *CV,* but this increases the cost of testing. Increasing *T* above 101.5% can also increase the maximum *CV*, but this can only be done with drug stability justification. A better approach may be to tighten the process capability of *X*, which allows a higher *CV* and improves *CV* by optimizing the process. Approaches to improving *X* and *CV* are explored below by examining the factors that influence *X* and *CV*, as well as their overall effect on *AV*.

### 3.1. Factors Affecting Accuracy (X)

Based on a component material balance on APIs for a drug-layering process, the number of variables determining the accuracy of the coating process was found to be dependent on the operating strategy [14]. When a certain quantity of coating solution containing an amorphous API is sprayed onto a fixed quantity of tablets, *X* is given by: (10)$$X_{1}=\frac{{\dot{m}}_{sp}\tau }{T_{cl}\left(w_{B}/w_{Tab}\right)}\frac{w_{d}x_{p}\left(1-x_{m}\right)}{w_{d}+w_{e}+w_{s}-w_{l}}\eta$$
where ${\dot{m}}_{sp}$ is the spray rate; *T_cl_* is the drug dosage claim; *w_B_* and *w_Tab_* are the weights of batch and tablet, respectively; *x_p_* and *x_m_* are the purity and moisture content of the API, respectively; *w_d_*, *w_e_*, and *w_s_* are the weights of the API, excipients, and solvents added to the coating solution, respectively; *w_l_* is the evaporation loss of the solvent prior to spraying; *η* is the coating efficiency. *η* is defined as the total coating mass on the tablets divided by the total solids mass sprayed onto the bed of tablets. Other than *η*, the variables in Equation (10) are related to the known material and operational parameters. Many of the operational variables can be eliminated by changing the operation such that the total amount of solution containing the API corresponds to the desired amount on the tablets. In this case, the factors that influence *X* are given by
(11)$$X_{2}=\frac{w_{d}x_{p}\left(1-x_{m}\right)}{T_{cl}\left(w_{B}/w_{Tab}\right)}\eta$$

Since the variables on the right-hand side (RHS) of Equation (11) are independent, the variability of *X* ($\sigma_{X}^{2}$) can be determined by summing the individual variances, i.e.,
(12)$$\sigma_{X}^{2}=\sigma_{w_{d}}^{2}+\sigma_{x_{p}}^{2}+\sigma_{x_{m}}^{2}+\sigma_{w_{B}}^{2}+\sigma_{w_{Tab}}^{2}+\sigma_{\eta }^{2}$$

As an illustration, to achieve a process capability index (*C_pk_*) [15] of 1.33 for *X* that meets the level 1 *AV* criterion for an ideal case, where the variances on the RHS of this equation are equal, each *σ_i_* must be less than

0.15% for 98.5–101.5% specification limit on *X*0.51% for 95–105% specification limit on *X*.

Other values of *C_pk_* (corresponding to different associated defect rates) for the above scenario are shown in Figure 3.

It should be noted that the parameters on the RHS of Equation (12), other than *η*, are not unique to pan coating processes, as *σ_i_* less than 0.51% (based on the 95–105% specification limit) is required and routinely met in pharmaceutical manufacturing. The challenge for DCT processes lies in the control of *η* and meeting the tighter requirement of *σ_i_* based on the 98.5–101.5% specification limit that allows for the *CV* to be as high as 6%.

The coating efficiency (*η*) is a function of many process parameters and requires consistent setup and controls [8]. The parameters with the highest impact on coating efficiency are atomizing and pattern air flow rates, spray rates, and gun-to-bed distance [16,17,18]. By keeping the nozzle assembly and setup consistent and ensuring the calibration of flowmeters, the batch-to-batch variability of the coating efficiency is minimized. If insoluble API or excipients are in the coating solution, the size of insoluble particulates also affects the coating efficiency. For this reason, wet ball mills may be used prior to coating to homogenize the coating solutions.

### 3.2. Factors Affecting Precision (CV)

In addition to the coating time, the material, equipment, and process parameters should also be optimized to improve coating uniformity. This can be achieved by using literature models predicting *CV*. In this work, the analytical model developed by Choi et al. [7,8] is used since it has shown good agreement with literature data [4,19,20] and correlations [12,13]. The Choi model gives *CV* as a function of tablet diameter (*d_p_*), tablet shape (ψ), bed porosity (*ε*), pan speed ($\dot{\varphi }$), coating time (*τ*), bed height (*h*), pan diameter (*D*), and the characteristic size of the spray zone (Δ):(13)$$CV=\sqrt{\frac{d_{p}^{1/6}}{k_{n}{\dot{\varphi }}^{2/3}\tau }\frac{\left(\theta +\sin\theta \right)\left(2\theta_{\Delta }-\sin\left(2\theta_{\Delta }\right)\right)}{\left(2\theta -\sin\left(2\theta \right)\right)-\left(2\theta_{\Delta }-\sin\left(2\theta_{\Delta }\right)\right)}}$$
(14)$$\mathrm{where}\;\theta =\cos^{-1}\left(1-2\frac{h}{D}\right)$$
(15)$$\theta_{\Delta }=\cos^{-1}\left(1-2\frac{h-\Delta }{D}\right)$$
(16)$$\mathrm{and}\;\Delta =k_{\Delta }^{\prime }\frac{d_{v}\psi \varepsilon }{1-\varepsilon }$$
where Δ is a function of the spray coverage of the bed surface and the spray penetration into the bed. *k*_Δ_ is a proportionality factor, and *d_v_* is the volume-equivalent sphere diameter of the tablet. Varying *τ* should be the last resort to improve coating uniformity (i.e., to lower *CV*) since it increases operating costs. The design of the tablet geometry and coating formulation, coating pan geometry, and other process variables should be optimized first before considering longer coating times. However, batch-to-batch variability is expected to be low once formulation, equipment, and process parameters are fixed or given tight specifications. Consistency in the equipment set up of baffles, nozzle assembly, and location, and ensuring the calibration status of coating solutions, atomizing air, and pattern air flow rates, are critical for product quality.

Once the tablet formulation and pan equipment with optimized nozzle setup are selected, the highest pan speed with minimal tolerable tablet breakage and attrition should be used. If the coating time is excessive, the batch size can be reduced to improve the cycle time.

For a desired *CV*, Equation (13) can be used to optimize the production rate ($\dot{B}$) based on the coating time (*τ*), changeover time (*t_co_*), and batch size (*w_B_*), i.e.,
(17)$$\dot{B}=\frac{w_{B}}{\tau +t_{co}}$$

For fixed changeover times, larger batches have shorter changeover times per unit of product due to the lower frequency of changeovers. However, as indicated by Equation (13), larger batches take longer to coat to the same coating uniformity. Consequently, larger batches, shown in Figure 4 as % pan loading, will have higher production rates for longer changeover times, while smaller batches will have higher production rates for shorter changeover times.

### 3.3. Practical Implications

The incremental cost of reducing *CV* for DCT is high (especially at low *CV* values) because the coating time increases exponentially with a decreasing *CV*. Longer processing times mean a lower capacity and higher operating costs. There is potentially a major disconnect between achieving the requisite *CV* values and minimizing manufacturing costs. However, on-specification manufacturing can still be achieved at high *CV* values (e.g., at a *CV* of 4–6%) with shorter coating times by minimizing the variability in *X*, as discussed above. Targeting *CV* below 4–6% may be impractical for manufacturing drug-layered tablets due to long coating times.

#### 3.3.1. Performance and Cost Comparison to Pellet Systems

Pellet systems can offer a lower *CV* than pan coaters but are susceptible to a higher variability in *X*. The target *CV_i_* for each pellet can be high since, when combined into capsules or MUPS, the overall *CV* is given by
(18)$$CV=\frac{CV_{i}}{\sqrt{N_{u}}}$$
where *N_u_* is the number of pellets in a capsule and is given by
(19)$$N_{u}=\frac{6V_{cap}\left(1-\varepsilon \right)}{\pi d_{s}^{3}\left(1+x_{wg}\rho_{s}/\rho_{c}\right)}$$
where *V_cap_*, *ε*, *d_s_*, *x_wg_*, *ρ_s_*, and *ρ_c_* are the capsule fill volume, interstitial pellet porosity, pellet seed diameter, weight gain, and densities of the seed and coating, respectively. Figure 5 shows the results predicted by Equation (18) for the selected commercially available capsule sizes. A capsule or MUPS containing 100 pellets with a *CV* of 15% results in a capsule *CV* of 1.5%. The implication of this is that *CV* requirements can be easily met with pellet systems. However, since the pellets are filled into capsules or combined with other excipients to form MUPS, the extra materials (e.g., the capsule shell and filler material) and volume-based dosing can further contribute to weight variability. Segregation due to differences in material properties between pellets and fillers may also be significant. This weight variability is directly associated with the process variability of *X,* and the control of *X* is therefore expected to be the greater challenge for pellet systems. For this reason, checkweighers are often used post-encapsulation to reject off-weight capsules. This adds another unit operation, and rejections from this operation lower the yield, both of which increase the manufacturing cost. Furthermore, increasing the number of unit operations typically affects both the product development time and cost adversely.

The capital and operating costs of pellet systems are generally higher than the costs of tablet pan coaters. Pellet processing systems are typically fluid beds that pneumatically mix pellets, with the most widely used pellet coaters using Wurster technology. Fluid bed processors are physically larger than pan coaters for equivalent batch sizes, having more ancillary parts and controls, and require more space, all of which increase the capital cost. Pellet and tablet coaters can both use seed materials sourced from vendors, but pellet systems require additional unit operations after drug-layering to combine the pellets into the final dosage forms. A larger number of unit operations also require more work-in-progress (WIP) areas, hold time controls for intermediate products, and more transaction controls. These factors add to the capacity requirement, supply-chain complexity, labor cost, and ultimately, the capital and operating cost.

Pellet systems may also result in a low coating efficiency (62–85% [21]) compared to pan coaters (up to 99%) [18]. For drug-layering, this difference in coating efficiency can significantly impact the operating cost due to the high cost of APIs.

#### 3.3.2. Other Considerations

From a process perspective, conventional tablet film coating processes need to be adjusted for drug-layering processes. For example, the accuracy and precision in terms of inter-tablet distribution of APIs are the most critical quality attributes in drug-layering processes, while appearance and intra-tablet uniformity are not as critical. Hence, reducing the spray atomization energy and thereby producing larger droplets while maintaining the same spray distribution can improve both coating efficiency and uniformity [8], but it can also result in the appearance of defects such as orange peel [9]. However, the appearance defects may be acceptable for DCT, provided the *AV* criteria are met. Application of an aesthetic coating over DCT can further reduce appearance defects if needed.

## 4. Conclusions

The manufacture of oral dosage forms containing amorphous solids by the pan coating of tablets involves significantly fewer unit operations than the more widely used processes based on micronizing, spray drying, and hot-melt extrusion, followed by encapsulation. This results in pan-coaters having relatively lower capital and equipment costs as well as reduced set-up times and process complexities.

It was shown that the pan-coating of tablets can meet the harmonized USP/EP/JP acceptance value (*AV*) criteria. Analyses of the factors affecting accuracy and precision, the two parameters determining *AV*, indicate that the main difficulty in manufacturing drug-coated tablets is the practical limitations of coating precision. However, model analyses also show that the *AV* criteria can be met by implementing appropriate monitors and controls around materials, maintenance, and setup and operation of processing equipment.

## Figures and Tables

**Figure 1 pharmaceutics-14-00149-f001:**
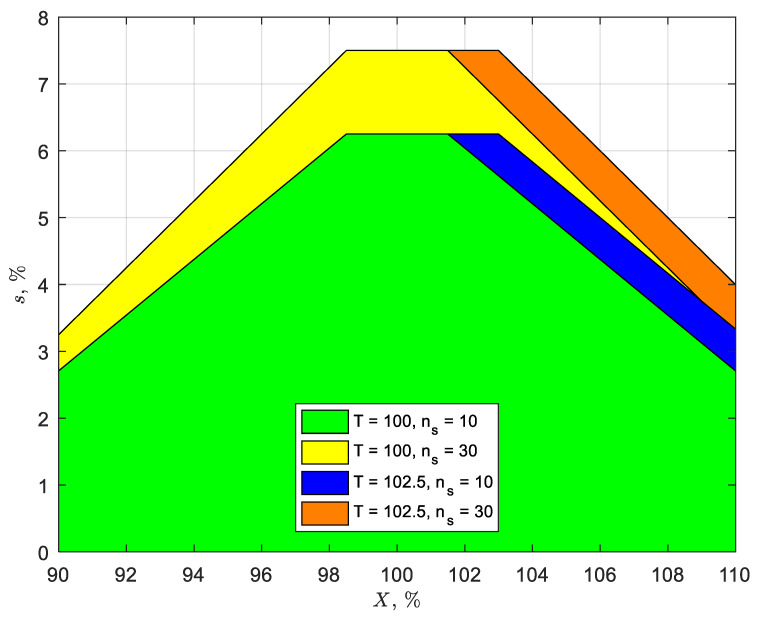
Acceptable ranges (i.e., *AV* ≤ 15) of the average (*X*) and standard deviation (*s*) for sample sizes (*n_s_*) of 10 and 30. Target content uniformity (*T*) = 100% (green and yellow areas) and 102.5% (blue and orange are the expanded areas).

**Figure 2 pharmaceutics-14-00149-f002:**
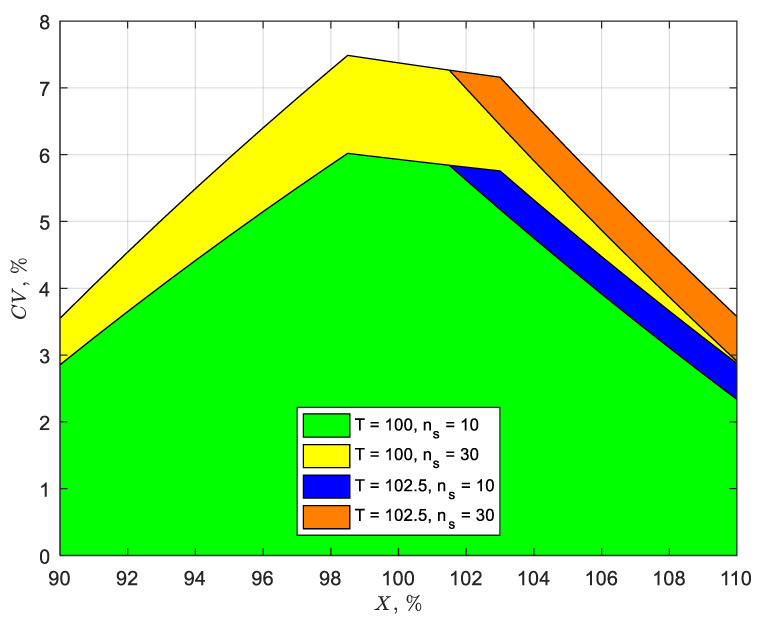
Acceptable ranges (i.e., *AV* ≤ 15) of the average (*X*) and coefficient of variation (*CV*) for samples sizes (*n_s_*) of 10 and 30. Target content uniformity (*T*) = 100% (green and yellow areas) and 102.5% (blue and orange are the expanded areas).

**Figure 3 pharmaceutics-14-00149-f003:**
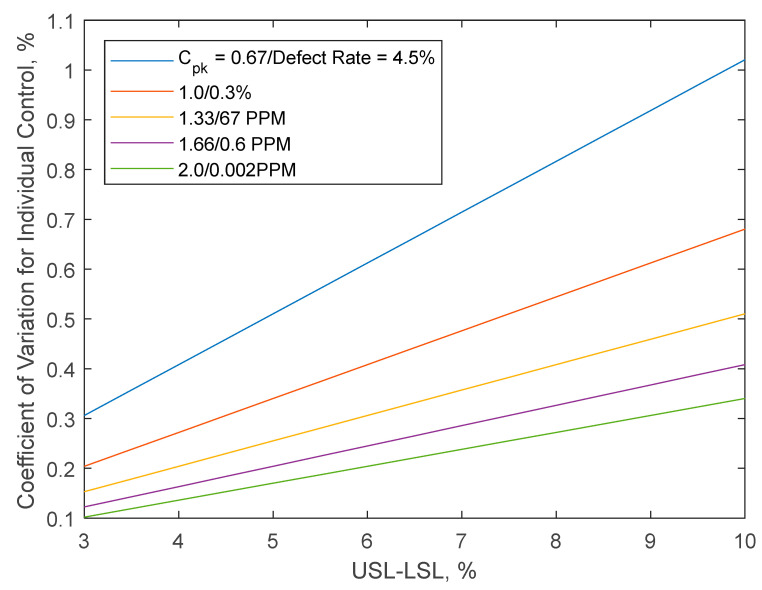
*CV* for individual controls (i.e., *σ_i_* on the RHS of Equation (12)) corresponds to *AV* = 15 at various *X* and *C_pk_* values. USL-LSL represents the difference in the upper and lower specification *X*.

**Figure 4 pharmaceutics-14-00149-f004:**
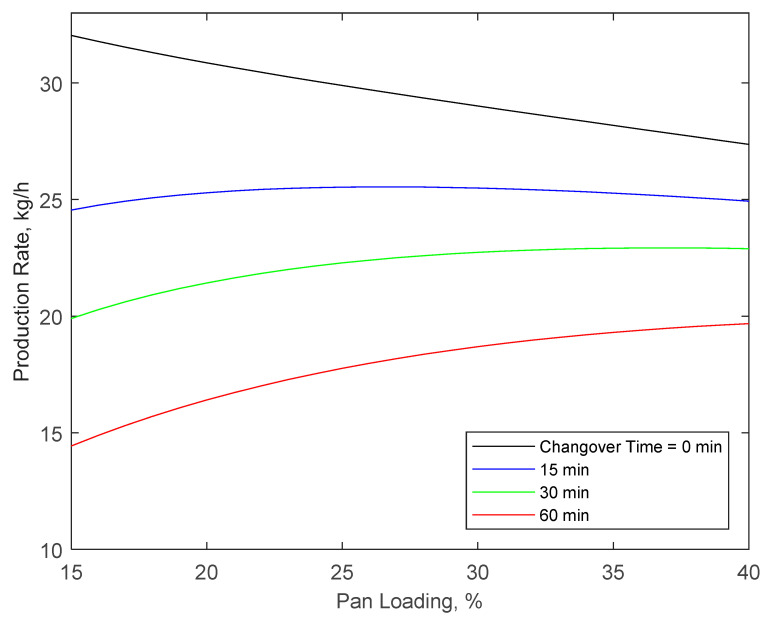
Production rates required to achieve *CV* = 5.8% as a function of batch size (given in pan loading) and changeover time between batches in 0.6 m diameter and 0.6 m length cylindrical pan.

**Figure 5 pharmaceutics-14-00149-f005:**
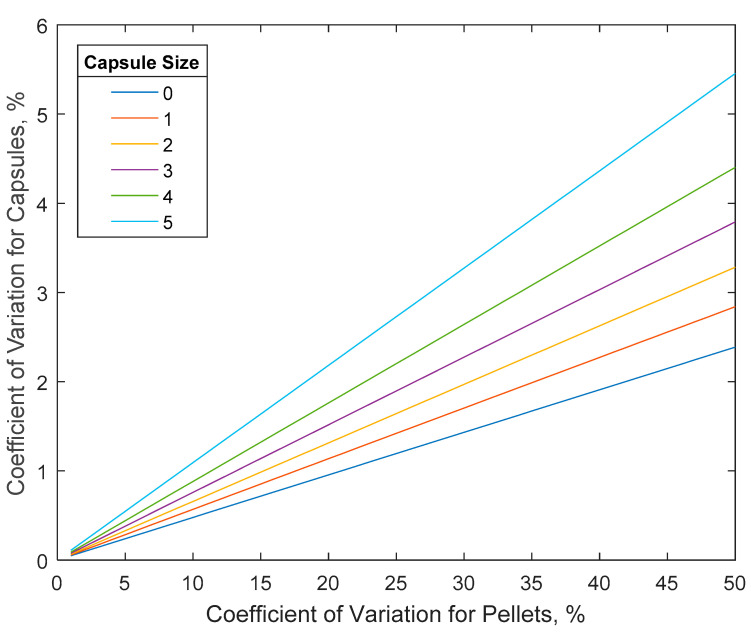
*CV* of capsules as a function of capsule size and *CV* of pellets for 1 mm seed pellet diameter, 100% coating weight gain, and seed and coating densities of 1.5 and 1.2 g mL^−1^, respectively.

**Table 1 pharmaceutics-14-00149-t001:** Principal unit operations ^†^ for the manufacture of amorphous APIs.

Unit Operations	Micronization of API	Solvent Spray Drying	Hot-Melt Extrusion	MUPS	Drug Layering onto Pellets	Drug Layering onto Tablets
Micronization	✓	✗	✗	✗	✗	✗
Blending	✓	✓	✓	✓	✗	✗
Spray Drying	✗	✓	✗	✗	✗	✗
Extrusion	✗	✗	✓	✗	✗	✗
Delumping/Milling	↔	↔	✓	✗	✗	✗
Tableting	✓	✓	✓	✓	✗	✗
Coating (Drug Layering or Other)	✓	✓	✓	✓	✓	✓
Encapsulation	✗	✗	✗	✗	✓	✗
Total Unit Ops	4 (5)	4 (5)	5	3	2	1

^†^ ‘Unit operations’ are defined as one or more types of equipment used to achieve the desired physical properties of a drug product. Symbols: ✓ typically used; ✗ not used; ↔ if required, total is shown in brackets.

## Data Availability

Not applicable.

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
