# Peer review of "Application of Mathematical Models to Determine the Feasibility of Amorphous Drug Layering in Pan Coaters"

_pharmaceutics, 2022, doi:10.3390/pharmaceutics14010149_

Round 1

Reviewer 1 Report

Issues addressed

Author Response

Thank you for your feedback.

Reviewer 2 Report

The manuscript described mathematical models of amorphous active pharmaceutical ingredient (API). The authors showed that pan-coating of tablets can meet harmonized acceptance value (AV) criteria. Thus, these findings will be useful for amorphous API manufacturing. Therefore, the manuscript is not too excellent to be published. In other words, the manuscript is so excellent that it should be published.

Comments

(1) Can engineered amorphous API on pellets or tablets be tolerable both to acidic condition in the stomach and basic condition in the small intestine?

(2) Can engineered amorphous API on pellets or tablets be released in the small intestine in basic condition?

(3) I am unfamiliar with the equations and the formulas. Are these formulas the standard methods to calculate values?

(4) Are the calculated values consistent with the observed ones?

That is all.

Author Response

Thank you so much for your review and questions.  Please find our answers to your questions below:

(1) Can engineered amorphous API on pellets or tablets be tolerable both to acidic condition in the stomach and basic condition in the small intestine?

  • Yes, either controlled-release or delayed-release coating can be applied over DCT to control the location and release rate of the API.  The description provided in the article regarding the uses of HPMCAS and/or HPMCP with amorphous API are examples of engineering solutions for release in the small intestine.

(2) Can engineered amorphous API on pellets or tablets be released in the small intestine in basic condition?

  • Yes, HPMCAS and HPMCP are example enteric coatings, which when combined with API or applied over the API layer would release the API in the small intestine.

(3) I am unfamiliar with the equations and the formulas. Are these formulas the standard methods to calculate values?

  • Many equations and formulas used are from our recent publications based on analytical derivations, which provide mathematical relationship and do not require computation-intensive stochastic or numerical approaches.  Some new relationships are also provided in this article.  Our aim is to demonstrate the utility of these mathematical relationships so that the critical factors can be better understood and controlled during formulation and process development, tech transfer, and manufacturing troubleshooting.

(4) Are the calculated values consistent with the observed ones?

  • These calculations were found to be in good agreement with the literature experimental results and literature models as referenced in this article (line #s 247-249).

This manuscript is a resubmission of an earlier submission. The following is a list of the peer review reports and author responses from that submission.

Round 1

Reviewer 1 Report

In this paper, the authors confirm that pan coating of tablets involves significantly fewer unit operations than micronizing, spray drying, and hot melt extrusion followed by encapsulation. In addition, they claimed that pan coating of tablets can meet standard USP/EP/JP acceptance value (AV) criteria. However, this paper not only lacks the verification experiment to support the conclusion, but also the introduction description lacks logic. For example, a) as a simple comparison, table 1 is not suitable for large space; b) what is more incomprehensible is the way of writing on page 2, 62-66.

Therefore, I think this paper is not appropriate for publication in Pharmaceutics.

Reviewer 2 Report

In this study, the authors address the use of pan coating technology in the preparation of oral solid dosage forms with amorphous solids. 
The authors used mathematical expressions (already proposed in the literature) to create 2D diagrams that theoretically describe product properties such as accuracy and precision. 
 e.g. the standard deviation of the sample compared to the mean individual API content for different values of sample size, etc. 

Main concerns:
What is new in this study? The authors have used some empirical mathematical functions to theoretically illustrate some properties of the formulations. However, I am trying to find out what this study adds to the scientific community in terms of novelty.

Minor:
Legends to Figures 1 and 2: Please define what the terms shown in the graph stand for.

Reviewer 3 Report

The manuscript described manufacturing of amorphous active pharmaceutical ingredient (API). The authors showed that pan-coating of tablets can meet standard USP/EP/JP acceptance value (AV) criteria. Thus, these findings will be useful for amorphous API manufacturing. Therefore, the manuscript is not too excellent to be published. In other words, the manuscript is so excellent that it should be published.

Comments

(1) Can engineered amorphous API be tolerable both to acidic condition in the stomach and basic condition in the small intestine? Are hydroxypropyl methylcellulose acetate succinate (HPMCAS) and hydroxypropyl methylcellulose phthalate (HPMCP) so?

(2) Can engineered amorphous API be released in the small intestine in basic condition?

(3) I am unfamiliar with the equations and the formulas. Are these formulas the standard methods to calculate values?

(4) Are the calculated values consistent with the observed ones?

That is all.

Reviewer 4 Report

The current manuscript presents a strong theoretical comparisons and industry-relevant analysis for creating oral solid dosage forms that contain amorphous APIs. The manuscript is extremely well presented and discussed and will serve as a valuable contribution to the Pharmaceutics community. I have no major comments and believed the manuscript can be accepted in current form. One minor comment is that the title for the manuscript is a little vague and could be improved, in my opinion.